# Stress and substance abuse among workers during the COVID-19 pandemic in an intensive care unit: A cross-sectional study

**Diego Vinicius Santinelli Pestana**[ID]*, **Dante Raglione**[ID], **Luiz Dalfior Junior, Caroline de Souza Pereira Liberatti, Elisangela Camargo Braga, Vitor Augusto de Lima Ezequiel, Adriana da Silva Alves, Juliana Gil Mauro, José Omar de Araújo Dias, Paulo Thadeu Fantinato Moreira, Bruno Del Bianco Madureira, Lilian Petroni Paiva, Bruno Melo Nóbrega de Lucena, João Manoel Silva Junior, Luiz Marcelo Sá Malbouisson**

Department of Anesthesiology, Intensive Care Unit, Instituto do Câncer do Estado de São Paulo/Hospital das Clínicas da Faculdade de Medicina da Universidade de São Paulo, São Paulo, Brazil

* diego.pestana@fm.usp.br

## Abstract

### Objective

Professionals working in intensive care units (ICUs) during the COVID-19 pandemic have been exposed to stressful situations and increased workload. The association between symptoms of traumatic stress disorders, substance abuse and personal/occupational characteristics of Brazilian COVID-19-ICU workers is still to be addressed. Our aim was to evaluate the prevalence of those conditions and to find if those associations exist.

### Methods

In this observational, single-center, cross-sectional study, all professionals working in a COVID-19 ICU were invited to fill an anonymous form containing screening tools for traumatic stress disorders and substance abuse, and a section with questions regarding personal and occupational information.

### Results

Three hundred seventy-six ICU professionals participated. Direct exposure to patients infected by COVID-19, history of relatives infected by COVID-19, and sex (female) were significantly associated with signs and symptoms of traumatic stress disorders. 76.5% of the participants had scores compatible with a diagnosis of traumatic stress disorders. Moreover, the prevalence of scores suggestive of Tobacco and Alcohol abuse were 11.7% and 24.7%, respectively.

### Conclusion

ICU workers had significantly elevated scores on both screening forms. Providing psychosocial support to ICU professionals may prevent future problems with traumatic stress disorders or substance abuse.

**Data Availability Statement:** All relevant data are within the paper and its Supporting Information files.

**Funding:** The author(s) received no specific funding for this work.

**Competing interests:** The authors have declared that no competing interests exist.

# Introduction

As the COVID-19 pandemic progresses, several hospitals have gone through changes in their usual work dynamics, among which are the elevated demand for intensive care unit (ICU) beds, the increased death rate, and preventive measures to avoid infection during healthcare. All of those elements are present in the daily routine of the ICU staff, interfering with their mental and physical health. Those factors may lead to increased workload and stress, therefore enhancing the risk of developing acute or chronic psychological disturbances (e.g., burnout, substance abuse [SA], acute stress disorder [ASD], and post-traumatic stress disorder [PTSD]) [1–4].

Even in non-pandemic periods, healthcare professionals working in ICUs suffer high stress loads possibly due to the environment and the severity of patients [5]. In this context, recurrent stressful stimuli may lead the professionals developing SA [6, 7] and ASD [8], more often than in other areas of healthcare [9–11]. Additionally, recent studies suggest those workers have been experiencing a high impact in their mental health during the pandemic [12, 13]. However, limited research has been conducted on the occurrence of those phenomena in healthcare professionals working in Brazilian ICUs during the COVID-19 pandemic. Furthermore, there is no data providing an independent analysis of the eventual associations between each single stressor element that may be present in the ICU environment and the occurrence of SA, PTSD, and ASD in that population [14].

Therefore, the aim of this study was to evaluate the occurrence of SA, PTSD, and ASD in professionals working in Brazilian ICUs, either having direct contact with patients or not, using validated questionnaires as tools to assess the prevalence of those conditions. The secondary objective was to identify associations between epidemiological and labor characteristics of those workers and the development of those psychological disturbances in that population.

# Methods

## Study design

This was an observational, single-center, cross-sectional, online questionnaire-based study. This study was carried out in an intensive care unit in an oncological hospital in São Paulo, Brazil.

## Study participants

All the professionals working in an oncological COVID-19 ICU, regardless of their role (health professionals and non-health professionals), who were on duty for at least during one shift through the period of July to October/2020, were invited to participate in this study. Exclusion criteria were as follows: age under 18, refusal to participate or to sign written informed consent, and being part of the group that contributed to the organization of this study. Additionally, those who were absent from work during the data collection period were excluded, regardless of the reason for absence (e.g., vacations, medical leave). All participants received a standardized approach. The study was approved by institutional ethics committees. All participants filled the questionnaires after signing the consent form during one of their work shifts, without interruptions. This project followed the guidelines of the Declaration of Helsinki.

**Approach to participants.** To avoid constraints and exposure, all participants were approached and invited to participate individually. A reserved, isolated room was always offered, and the participants could use it while filling the online forms. Participants from all shifts and sections working in the COVID-19 ICU were invited. All answers were kept private and anonymous, and were obtained in one visit.

## Measurements

Participants filled a standardized form containing personal and professional information (see Supporting Information section); they also answered screening forms for SA (Alcohol Smoking and Substance Involvement Screening Test [ASSIST 2.0]) [15, 16], ASD, and PTSD (using the Impact of Events Scale-Revised [IES-R]) [17, 18]. The IES-R scores were categorized as follows: 1–11 = few/no signs of ASD/PTSD; 12–32 = several signs of ASD/PTSD, patient monitoring is recommended; ≥33 = highly suggestive of ASD/PTSD, immediate psychiatric evaluation is recommended [17]. The ASSIST 2.0 scores were categorized as follows: 0–3 = occasional substance use; 4–15 = suggestive of substance abuse; ≥16 = suggestive of substance dependence [15]. The ASSIST 2.0 provided separate scores for each substance category. Frequencies were calculated and associations between the participants characteristics and the results of the questionnaires were made. Questionnaires were made available as Supporting Information S1–S6 Files.

## Statistical analysis

Categorical variables were expressed in frequencies. The Quantitative variables were classified according to normal or non-normal distribution using Shapiro-Wilk's test and were expressed with mean and standard deviation or median and interquartile range. The means of continuous variables with normal distribution were compared using Student T test for independent variables. The medians of continuous variables with non-normal distribution were compared using Mann-Whitney's U test. Categorical variables were compared using Chi-square test or Fisher's exact test when indicated. When appropriate, multivariate logistic regression analyses were used to explore the relationship among variables, calculating measures of association among variables expressed by odds ratio and a 95% confidence interval. Moreover, the regression analyses were used to adjust eventual confounding factors given the variable number of participants according to different specialties and roles in the ICU. All tests were two-sided. Values of $p < 0.05$ were considered statistically significant. All statistical analyses were performed using STATA/MP Version, 16.1 (for MAC) (StataCorp. 2019. Stata Statistical Software: Release 16. College Station, TX: StataCorp LLC).

## Results

Three hundred seventy-six ICU professionals agreed to participate in this survey, 340 with direct exposure to COVID-19 infected patients (see Table 1). Professionals directly exposed to infected patients included 54 physicians (14%), 53 physiotherapists (14%), 226 nurses/technicians (60%), 38 pharmacists (10%), 5 nutritionists (1%), 7 psychologists (1.5%) and 3 social service professionals (0.7%). 71.5% of the participants were female. The age variable was categorized, and participants who were less than 38 years of age were the most frequent (56.4%). More than 80% of the workers who were directly exposed participated in this study. It was not possible to determine the answer ratio among workers without direct exposure as most of them (e.g., security, cleaning services) were employed by third-party companies. Less than ten individuals refused to participate.

Most participants (66.5%) had a weekly exposure greater than 24 hours. 50% of the participants had been infected by COVID-19, and 38% of them had had a relative infected. Although most of them shared their homes with individuals from vulnerable populations (e.g., elderly relatives), only 21% of them sought alternative accommodation away from home during the pandemic. Forty-three participants (11.5%) declared to have a history of psychiatric disease. Table 1 describes the characteristics of the studied individuals.

**Table 1. Socio demographic, personal, and occupational characteristics of participants.**

| Baseline characteristic | ICU professionals | |
|---|---|---|
| | *n* | *%* |
| Gender | | |
| Female | 269 | 71.5 |
| Male | 107 | 28.5 |
| Age[*] | | |
| > 38 years of age | 161 | 43.6 |
| < 38 years of age | 215 | 56.4 |
| Occupational exposure to infected patients (exposed) | 340 | 90.6 |
| Time exposed to infected patients | | |
| Non-exposed | 39 | 10.4 |
| Less than 24h/week | 87 | 23.1 |
| Over 24h/week | 250 | 66.5 |
| History of psychiatric diseases[a] | 43 | 11.5 |
| Infected by COVID-19[a] | 187 | 50.0 |
| Relative infected by COVID-19[a] | 143 | 38.0 |
| Living with high-risk group relatives for COVID-19[a] | 240 | 63.8 |
| Sought alternative accommodation away from home[a] | 80 | 21.3 |

*Note.* N = 376 (total number of participants); n = number of participants within the subgroup; ICU = Intensive Care Unit.

[*] Age groups were categorized and ages <38 years of age were the most frequent.

[a] Reflects the number and percentage of participants answering "yes" to this question.

## IES-R score results

Most participants (76.3%) had scores ≥12 (see Table 2). One-hundred twenty-seven (33.8%) of them had scores higher than 33, which strongly suggests a diagnosis of PTSD and the need of psychiatric evaluation as soon as possible. Eighty-nine (23.7%) participants were free from PTSD signs or only had a few of them. Results are summarized on Table 2.

 **Psychiatric medical history.** The relationship between IES-R scores and history of psychiatric disease was assessed, but those factors did not present a statistically significant association (p = 0.341, OR = 2.68 [95% CI = 0.35–20.51]) (see Table 3). Then, the relationship between IES-R scores and history of anxiety disorders (which included ASD and PTSD) was

**Table 2. Frequencies of IES-R score categories.**

| Score Categories | ICU professionals | |
|---|---|---|
| | *n* | *%* |
| 1–11 | 89 | 23.7 |
| 12–32 | 160 | 42.5 |
| ≥33 | 127 | 33.8 |

*Note.* N = 376 (total number of participants); n = number of participants within the subgroup.

Mean IES-R score = 26.6, Standard Deviation = 17.9

IES-R = Impact of Event Scale–Revised; scores were categorized according to the clinical interpretation of the results.

[*] Categories: 1–11 = few/no signs of traumatic stress disorder, 12–32 = several signs of traumatic disorder, patient follow up is recommended, >33 = high probability of ongoing traumatic stress disorder, immediate psychiatric evaluation is recommended [17].

**Table 3. Factors without an association with IES-R scores.**

| Logistic parameter | IES-R scores | | |
|---|---|---|---|
| | *OR* | *CI* | *p* |
| History of psychiatric disease | 2.68 | 0.35–20.51 | 0.341 |
| History of anxiety disorders | 2.23 | 0.29–17.13 | 0.44 |
| Sought alternative accommodation away from home | 5.72 | 0.75–43.32 | 0.091 |

*Note*. OR = Odds Ratio, CI = 95% confidence interval, IES-R = Impact of Event Scale–Revised.

assessed, but those factors also did not present a statistically significant association (p = 0.44, OR = 2.23 [95% CI = 0.29–17.13]). Table 3 summarizes those findings.

**Personal and occupational factors.** Among personal factors, being a female and having a relative infected by COVID-19 were associated with higher IES-R scores (Table 4). When compared to men, women had a higher median score (27 vs 19 in males [p = 0.0001]) even after adjustment with Wilcoxon rank sum test. Professionals with infected relatives (OR = 3.91 [p = 0.031, 95% CI = 1.13–13.50]) were also significantly more likely to score higher in the IES-R, therefore more likely to suffer from PTSD or ASD (Table 4). The relationship between those who had sought alternative accommodation away from home during the pandemic in order to protect their relatives and IES-R scores was also assessed, but it was not statistically significant (OR = 5.72 [p = 0.091, 95% CI = 0.75–43.32])—see Table 3 above.

Among occupational factors, direct exposure to infected patients was associated with higher IES-R scores. When compared to professionals who were not directly exposed, those who had a direct exposure were more likely to achieve higher scores, with an OR = 5.62 (p = 0.001, 95% CI = 2.10–15.03). Those findings are presented in Table 4.

## Associations with substance abuse

Results indicate that, according to the ASSIST 2.0 form, 24.7% of ICU workers had scores suggestive of alcohol abuse. Scores compatible with tobacco abuse and dependence were seen in 11.2% and 2.3%, respectively. Eighteen workers (4.8%) had scores compatible with hypnotics abuse.

There was no statistically significant association between personal or occupational factors and substance use. However, two association trends were found. For tobacco and alcohol, higher scores had an association trend towards history of psychiatric disease, direct exposure to infected patients and those who sought alternative accommodation away from home during the pandemic in order to protect their relatives.

**Table 4. Results of multivariate logistic regression of factors associated with increased IES-R scores.**

| Logistic parameter | | IES-R scores | | |
|---|---|---|---|---|
| | *M* | *OR* | *CI* | *p* |
| Sex (female) | 27 | - | - | 0.0001 |
| Direct exposure to infected patients | - | 5.62 | 2.10–15.03 | 0.001 |
| Relative infected by COVID-19 | - | 3.91 | 1.13–13.50 | 0.031 |

*Note*. M = Median IES-R score, OR = Odds Ratio, CI = 95% confidence interval, IES-R = Impact of Event Scale–Revised.

IES-R scores of Female and Male participants were compared and then adjusted using Wilcoxon rank sum test (Male Median = 19).

**Table 5. Prevalence of substance use according to the ASSIST 2.0 categories.**

| Baseline characteristic | Suggestive of Substance Abuse | | Suggestive of Substance Dependence | |
|---|---|---|---|---|
| | *n* | *%* | *n* | *%* |
| Tobacco | 42 | 11.2 | 8 | 2.1 |
| Alcohol | 93 | 24.7 | 0 | 0 |
| Cannabis | 5 | 1.3 | 0 | 0 |
| Cocaine | 0 | 0 | 0 | 0 |
| Stimulants | 2 | 0.5 | 0 | 0 |
| Inhaled drugs | 0 | 0 | 0 | 0 |
| Hypnotics | 18 | 4.8 | 1 | 0.3 |
| Hallucinogens | 1 | 0.3 | 0 | 0 |
| Opioids | 2 | 0.5 | 0 | 0 |
| Intravenous drugs | 3 | 0.8 | 0 | 0 |
| Other drugs | 1 | 0.3 | 0 | 0 |

*Note*. $N$ = 376 (total number of participants); n = number of participants within the subgroup.

ASSIST 2.0 = Alcohol, Smoking and Substance Involvement Screening Test (version 2.0); scores were categorized according to the clinical interpretation of the results.

*Categories: 0–3 = occasional substance use; 4–15 = suggestive of substance abuse; ≥16 = suggestive of substance dependence [15].

For cannabis, stimulants, and cocaine, higher scores had an association trend towards history of psychiatric disease and direct exposure to infected patients. The prevalence of each substance use is detailed on Table 5 below.

## Discussion

This research found associations between participants' personal and occupational factors and signs of ASD/PTSD in a COVID-19 ICU in this Brazilian hospital. Sex (female), direct exposure to infected patients and having a relative infected by COVID-19 were significantly associated with higher IES-R scores. Other factors such as history of psychiatric disease, history of anxiety disorders, and seeking alternative accommodation away from home during the pandemic were assessed but did not present a statistically significant association with higher IES-R scores. The prevalence of alcohol and tobacco abuse was high. The evaluation of substance use among ICU workers with the ASSIST 2.0 form was not able to determine any statistically significant associations with personal or occupational factors and higher scores. Nonetheless, there was an association trend towards tobacco, alcohol, cannabis, stimulants, and cocaine use in those with a history of psychiatric disease and with direct exposure to infected patients. In a recent study evaluating the mental health of ICU workers in England [19], authors found similar results, with high prevalence of traumatic stress disorders and alcohol abuse, especially among nurses (which account for the most numerous subpopulation of the presenting study).

We believe that those factors associated with higher IES-R scores were directly related to an increased stress load in those individuals. Whilst the cause is unclear, it could be suggested that, when compared to most of the occupational factors, personal life characteristics, as well as direct exposure to infected patients, were more strongly related to the likelihood of having PTSD or ASD symptoms. Researchers investigating the impact of personal and occupational factors on the mental health of Canadian ICU workers exposed to COVID-19 infected patients found that, following a multivariate analysis, those elements related to the anxiety about being infected during work were the best predictors of scores indicative of traumatic stress disorders

[20]. We further hypothesize that worrying about the safety of their relatives is another possible cause of this overwhelming stress.

In terms of substance use, the most frequently consumed were tobacco and alcohol. Those were also the substances with higher frequencies of abuse and dependence. It is possible that those results reflected the availability of those substances. Also, the abusive consumption of those substances could be a coping mechanism during the pandemic. Even considering the socio-cultural differences among countries, the current results are in accordance with those presented by other authors, in other populations of ICU workers (e.g., Netherlands [4], Canada [20], United States [21], and England [19]). Thus, it is likely that the findings of the presenting study could be generalizable to ICU workers from different populations and should be taken into consideration by professionals working globally.

This study has several limitations. Firstly, as reported and studied in the field of substance abuse, this study may be subject to social desirability bias, which is the tendency of a participant providing answers based on what is considered a desired behavior within a specific socio-cultural environment [22, 23]. The fear of being treated with prejudice by their peers may also be a contributing factor for this bias in ICU workers [24]. Secondly, some ICU staff during the application of this study either refused to participate or were on medical leave. Given that some of those workers may have had psychological or psychiatric reasons for work leave, our findings can be underestimated. Thirdly, the lack of statistical significance could be due to a small sample size in some sub-groups and thus statistical testing lacked power. However, with regards to the non-statistically significant results, we believe they should be taken into account because it is reasonable to hypothesize that some of those factors may also contribute to the psychological burden in the ICU workers. Fourthly, this study had no follow-up sessions. Therefore, it was not possible to determine causal relationships or risk factors. Finally, it was a single-centered research and, therefore, influenced by local environmental biases. Nevertheless, our sample was large enough to find some statistically significant associations between participants characteristics and the development of PTSD/ASD symptoms during the COVID-19 pandemic. It was also possible to highlight the alarming prevalence of substance use among those individuals, which may carry devastating consequences for their personal and professional lives.

Considering the duration and the consequences of the COVID-19 pandemic, specifically for the population of this study, our results should bring to light the need for institutional support for those individuals. From the experiences with the severe acute respiratory syndrome virus (SARS) outbreak in the early 2000's, it has been demonstrated that ICU professionals may need long term support for mental health issues developed during their work time with infected patients [25]. As recently proposed by other authors [26, 27], intervention plans should be traced not only during the pandemic but also after it, due to the prolonged course of diseases such as PTSD, ASD, and SA. For that reason, we believe that this study could be used for guiding larger, multicenter studies and institutional interventions to prevent and reduce the suffering of professionals dealing with similar situations in Brazil and in other countries.

## Supporting information

**S1 File. Personal and occupational information (English version).**
(DOCX)

**S2 File. Personal and occupational information (Portuguese version).**
(DOCX)

**S3 File. ASSIST 2.0 (English version).**
(DOCX)

**S4 File. ASSIST 2.0 (Portuguese version).**
(DOCX)

**S5 File. IES-R (English version).**
(DOCX)

**S6 File. IES-R (Portuguese version).**
(DOCX)

**S7 File. Minimal anonymized data set (English [translation]).**
(XLSX)

**S8 File. Minimal anonymized data set (Portuguese [original]).**
(XLSX)

## Author Contributions

**Conceptualization:** Diego Vinicius Santinelli Pestana, Dante Raglione, Luiz Dalfior Junior.

**Data curation:** Diego Vinicius Santinelli Pestana, Caroline de Souza Pereira Liberatti, Elisangela Camargo Braga, Vitor Augusto de Lima Ezequiel, Adriana da Silva Alves, Juliana Gil Mauro, José Omar de Araújo Dias, Paulo Thadeu Fantinato Moreira, Bruno Del Bianco Madureira, Lilian Petroni Paiva, Bruno Melo Nóbrega de Lucena.

**Formal analysis:** Dante Raglione, Luiz Dalfior Junior.

**Investigation:** Caroline de Souza Pereira Liberatti, Elisangela Camargo Braga, Vitor Augusto de Lima Ezequiel, Adriana da Silva Alves, Juliana Gil Mauro, José Omar de Araújo Dias, Paulo Thadeu Fantinato Moreira, Bruno Del Bianco Madureira, Lilian Petroni Paiva, Bruno Melo Nóbrega de Lucena.

**Methodology:** Diego Vinicius Santinelli Pestana, Dante Raglione, Luiz Dalfior Junior.

**Project administration:** Diego Vinicius Santinelli Pestana, Caroline de Souza Pereira Liberatti, Elisangela Camargo Braga.

**Supervision:** Diego Vinicius Santinelli Pestana, Dante Raglione.

**Visualization:** Diego Vinicius Santinelli Pestana, João Manoel Silva Junior, Luiz Marcelo Sá Malbouisson.

**Writing – original draft:** Diego Vinicius Santinelli Pestana, Dante Raglione, Luiz Dalfior Junior.

**Writing – review & editing:** Diego Vinicius Santinelli Pestana, Dante Raglione, João Manoel Silva Junior, Luiz Marcelo Sá Malbouisson.

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
