## [Decision Letter · Decision Letter 0]

15 Oct 2021

PONE-D-21-17265

Stress and Substance Abuse among workers during the COVID-19 Pandemic in an Intensive Care Unit: a cross-sectional study.

PLOS ONE

Dear Dr. Pestana,

Thank you for submitting your manuscript to PLOS ONE. After careful consideration, we feel that it has merit but does not fully meet PLOS ONE’s publication criteria as it currently stands. Therefore, we invite you to submit a revised version of the manuscript that addresses the points raised during the review process.

The manuscript has been evaluated by three reviewers, and their comments are available below.

The reviewers have raised a number of concerns that need attention, and they request additional information on methodological aspects of the study and the analyses, and they have requested additional contextualization and revisions to the language usage.

Could you please revise the manuscript to carefully address the concerns raised?

We look forward to receiving your revised manuscript.

Kind regards,

Vanessa Carels

Staff Editor

PLOS ONE

Additional Editor Comments (if provided):

Reviewers' comments:

Reviewer's Responses to Questions

**Comments to the Author**

1. Is the manuscript technically sound, and do the data support the conclusions?

Reviewer #1: Yes

Reviewer #2: Yes

Reviewer #3: Yes

2. Has the statistical analysis been performed appropriately and rigorously? 

Reviewer #1: Yes

Reviewer #2: No

Reviewer #3: Yes

3. Have the authors made all data underlying the findings in their manuscript fully available?

Reviewer #1: No

Reviewer #2: No

Reviewer #3: Yes

4. Is the manuscript presented in an intelligible fashion and written in standard English?

Reviewer #1: No

Reviewer #2: Yes

Reviewer #3: Yes

5. Review Comments to the Author

Reviewer #1: The study provides findings on an important topic that is of concern during this pandemic. Nonetheless, there are several major amendments that will need to be made before it can be considered for publication.

1. I am not certain that the claim the authors make about how there are only a few research studies that have examined the occurrence of mental health problems in healthcare workers working in the ICU during the pandemic is valid. From what I understand, this topic has garnered a lot of interest and there is substantial literature that has emerged since the pandemic. Perhaps it may be more appropriate to limit this statement to the context of Brazil.

2. Although the discussion does follow from the findings of the analysis, there is a lack of reference to existing literature about the topic. Authors should attempt to use existing literature to support their conclusions, or to posit reasons for their findings. Similarly, the limitations section suggests that participants may have been afraid of suffering retaliation, leading to report bias. Reasons for this should be provided in greater detail, or citations for studies that have looked into this phenomenon should be given. Lastly, the section on practical implications of this study should have references to studies that have highlighted the importance of institutional support and/or long term intervention plans.

3. The appropriate statistical methods were chosen.

4. The authors are unable to make the data publicly available due to its sensitive nature, but it is available upon reasonable request.

5. The language errors make it difficult to understand at times, and the awkward expression of certain terms may cause readers to misunderstand what the authors are trying to convey.

e.g "licensed from work" -- I believe the authors were trying to say that these healthcare workers were absent from work?

"Although most of them shared their homes with individuals with greater risk for severe infection, only 21% of them left home during the pandemic" -- It is unclear why or how these individuals are at greater risk of severe infection. Do you mean to say they belong to a vulnerable population (e.g chronic illnesses/elderly)? The term "left home" should be more appropriately expressed as "sought alternative accommodation away from home".

"However, with regards to the non-statistically significant associations, we believe they should be taken into account because of the biological plausibility they carry within them." -- Am not sure what biological plausibility means in this context

"Pharmaceuticals" -- I believe it should be "pharmacists"

6. Other comments:

- The abstract should not contain abbreviations that have not been spelled out in full at first mention.

- Please provide a citation for the way the IES-R and ASSIST 2.0 scores were categorized. Are there scoring guidelines available?

- Please refer to a standardized format (e.g APA) for the presentation of data in tables. The way it is currently organized can be confusing for readers. Also, all abbreviations in tables and its title (e.g ICU, IES-R) should be explained in the footnotes.

Reviewer #2: The study is interesting but is single-centre. There is no comparison between the different critical care professions.

The authors do not report anxiety or depression scores.

Finally, the authors did not go into enough depth in their analyses.

Reviewer #3: The title is accurate or relevant

The aims of the study are clearly stated

The study is original

The study is useful and relevant to the aims of the Journal

The design of the study is appropriate

The sample size, selection and composition are appropriate

Methods used to collect data (e.g. validated questionnaires and instruments, observational techniques) are appropriate

Qualitative or quantitative methods used to analyse the data are appropriate

Details of the methods (including settings and locations, procedures, dates of recruitment and follow-up or main outcomes) are clearly reported

The data are less than 5 years old

The study was approved by a research ethics committee prior to data collection

Participants were asked for informed consent prior to data collection or informed consent was not required

The qualitative or quantitative analyses were applied appropriately

Missing data, e.g. non-respondents, drop-outs or non-responses, have been accounted for

The results are clearly presented and explained

No further qualitative or quantitative analysis is required

The authors reflect on the strengths and limitations of the study

The results are compared to related findings in the literature

The results are discussed in relation to the relevant research, practice or policy issues

The discussion and conclusions do not speculate beyond what has been shown in this study

The article has a logical construction in a suitable format

The article has an appropriate length (not unnecessarily long or too short to be useful)

The writing is in a good standard of English, grammatically correct and easy to understand

The abstract is in an unstructured format and is sufficiently informative

Any tables and figures are all necessary, clearly annotated and easy to follow

6. PLOS authors have the option to publish the peer review history of their article (what does this mean?). If published, this will include your full peer review and any attached files.

Reviewer #1: No

Reviewer #2: No

Reviewer #3: **Yes: **Modesto Leite Rolim Neto - Faculdade de Medicina - Universidade Federal do Cariri - UFCA

---

## [Author Response · Author response to Decision Letter 0]

27 Nov 2021

Response to the Editor and Reviewers

Dear Reviewers and Dr. Carels (Editorial Board of PLOS One). We are pleased to share with you our response to the comments made regarding our manuscript entitled “Stress and Substance Abuse among workers during the COVID-19 Pandemic in an Intensive Care Unit: a cross-sectional study”. We hope that our replies can properly address the issues highlighted by you. Thank you for your attention to this.

Sincerely,

Diego V. S. Pestana (on behalf of the authors)

Comments – Editor:

Author’s Reply: Dear Dr. Carels, thank you for your help during this process and for your important comments. The manuscript has been edited to meet PLOS ONE’s requirements. We conducted a thorough review of the requirements and we believe that the manuscript now completely meets the publishing requirements. Also, we reviewed the text once more in search for eventual typos or misspellings that may have been unnoticed so far. Finally, as mentioned below in one of the replies, the text has been reviewed by 2 independent native English speakers to improve the quality the written English used on it and facilitate readers understanding. Nevertheless, if you or the reviewers happen to have any other suggestion regarding this topic, we will be glad to work on it.

Author’s Reply: We apologize for the inconvenience regarding this topic. As suggested, we uploaded 6 files as Supporting Information, accounting for each one of the components of the questionnaire used in this project (the ASSIST 2.0, the IES-R, and the questions regarding personal and occupational information developed by our research group) and both, an English and a Portuguese version of them.

3 & 4

Author’s Reply: Thank you for your guidance over this topic. As mentioned, we had not submitted the data set along with the other files given it involves information of sensitive nature. Nevertheless, after carefully reviewing PLOS ONE’s data policy, we believe that it is reasonable to share the data set, as it has no legal or ethical restrictions and can be publicly available as the minimal anonymized data set. In that way, data can be safely shared with readers without harming participants privacy.

Comments - Reviewer #1

The study provides findings on an important topic that is of concern during this pandemic. Nonetheless, there are several major amendments that will need to be made before it can be considered for publication.

1. I am not certain that the claim the authors make about how there are only a few research studies that have examined the occurrence of mental health problems in healthcare workers working in the ICU during the pandemic is valid. From what I understand, this topic has garnered a lot of interest and there is substantial literature that has emerged since the pandemic. Perhaps it may be more appropriate to limit this statement to the context of Brazil.

Author’s Reply: Dear Reviewer 1, thank you for your valuable comments! Indeed, within the last months several articles have been published, most of them exploring the consequences of the COVID-19 pandemic to the mental health of health care professionals working in ICUs. After carefully considering this comment, we conducted another search on literature, focusing on those evaluating Brazilian ICU workers. Our findings and conclusions are summarized below:

#1: Most of these articles approach mental health issues on Brazilian physicians and nursing staff. The present article evaluates not only physicians, but also all the nursing staff, physiotherapy professionals, nutritional professionals, cleaning staff, maintenance staff, administrative staff, and security staff. We believe that all of these professionals are essential to provide adequate conditions for an ICU, and, therefore, this would be an interesting feature of the present article.

#2: We used different searching methods and sources in order to identify other articles that evaluated the impact of COVID-19 on Brazilian ICU workers, specifically on the issue of substance abuse. Also, the presenting manuscript evaluated several personal and/or occupational characteristics rarely found in other studies of this nature However, we were not able to find any other study addressing these concerns in the studied population (Brazilian ICU workers). Therefore, we believe that this aspect would also provide scientifically relevant information to readers of PLOS One.

#3: We agree with your comment, and our manuscript has been properly changed to provide more accurate information. The reviewed version limits its claims to Brazilian ICU workers.

2. Although the discussion does follow from the findings of the analysis, there is a lack of reference to existing literature about the topic. Authors should attempt to use existing literature to support their conclusions, or to posit reasons for their findings. 

Similarly, the limitations section suggests that participants may have been afraid of suffering retaliation, leading to report bias. Reasons for this should be provided in greater detail, or citations for studies that have looked into this phenomenon should be given. 

Lastly, the section on practical implications of this study should have references to studies that have highlighted the importance of institutional support and/or long term intervention plans.

Author’s Reply: After carefully considering this comment, we reviewed our Discussion section. While reviewing it, we also searched for references that support the points highlighted in this comment. Below we addressed each point made in the comment above, using references that illustrate the topic under discussion.

#1: The concern about mental health among ICU workers during the COVID-19 pandemic has been widely studied within the last 2 years [1.1], [1.2], [1.3]. In a study with a structure similar to the one used in the present manuscript, with similar studied populations, Greenberg et al (2021) [1.4] identified alarmingly high prevalence of post-traumatic stress disorder symptoms, depressive symptoms, and signs of alcohol abuse among participants. Even before the COVID-19 pandemic, both substance abuse [1.5], [1.6] and traumatic stress disorders [1.7] were identified as being serious problems among healthcare professionals, often more intensely present in subspecialties such as Anesthesiology, Emergency Care, and Critical Care. Therefore, we believe that the findings reported in the present manuscript are in accordance with what could be expected, based on the pre-existing literature on this topic. Also, although we present novel scientific data, the pre-existing literature seems to support our findings and conclusions. We are thankful for this suggestion, and the Discussion section of the presenting manuscript has been properly adjusted after it.

#2: Substance abuse is a sensitive topic, often associated with stigmatization. Even in the context of healthcare, when it comes to individuals suffering with substance abuse, providers may act based on prejudice, as described by Stone et al (2021) [2.1]. This phenomenon may be associated with another factor known as “social desirability bias”, often described in self-report studies regarding substance abuse. According to Latkins et al (2017) [2.2]: “Social desirability bias is the tendency to underreport socially undesirable attitudes and behaviors and to over report more desirable attributes”. Early in the study of social desirability bias, Welte et al (1993) [2.3] concluded that “social desirability response bias probably results in underestimates of rates of heavy drinking and drug use” in self-reported studies. After reviewing these concepts and searching deeper into the literature, we believe that the Discussion section of our manuscript should approach the aforementioned sources of bias using the term “social desirability bias” instead of “report bias”. Accordingly, we re-wrote this section using the proper terminology and adjusting it to the proper phenomena described. Of note, there were no reports or any signs of retaliation during this study, and the incorporation of this element to the manuscript was a result of our misinterpretation of the involved biases.

#3: In terms of background, from the experiences with the 2003 SARS outbreaks, Maunder et al (2006) [3.1] demonstrated that healthcare providers are found to have an increased risk of developing mental health problems following a contagious disease pandemic. The awareness of this potential issue stimulated researchers worldwide to investigate the need for psychosocial support to healthcare workers, specifically those working at ICUs. Among recently published studies regarding the COVID-19 pandemic, Roberts et al (2021) [3.2] highlighted the importance of supporting ICU workers to prevent and/or diminish the harms involved with mental health issues in this population, claiming that "It is also clear that psychological support and services for nurses and the wider healthcare team need to be available and quickly convened in the event of similar major incidents, either global or local" [3.2]. Finally, the British Medical Journal published an article from their own authorship stressing the need for psychosocial support to these workers [3.3]. Those were only a couple of illustrative examples among several other articles our team was able to find. Therefore, we believe that it is possible to claim that other studies and/or authors also highlighted the importance of institutional support and/or long-term intervention plans.

We were happy to incorporate the aforementioned topics into the Discussion section of the presenting manuscript.

REFERENCES

[1.1] Kok, N., van Gurp, J., Teerenstra, S., van der Hoeven, H., Fuchs, M., Hoedemaekers, C., & Zegers, M. (2021). Coronavirus Disease 2019 Immediately Increases Burnout Symptoms in ICU Professionals: A Longitudinal Cohort Study. Critical care medicine, 49(3), 419–427. https://doi.org/10.1097/CCM.0000000000004865

[1.2] Binnie, A., Moura, K., Moura, C., D'Aragon, F., & Tsang, J. (2021). Psychosocial distress amongst Canadian intensive care unit healthcare workers during the acceleration phase of the COVID-19 pandemic. PloS one, 16(8), e0254708. https://doi.org/10.1371/journal.pone.0254708

[1.3] Danet Danet A. (2021). Psychological impact of COVID-19 pandemic in Western frontline healthcare professionals. A systematic review. Impacto psicológico de la COVID-19 en profesionales sanitarios de primera línea en el ámbito occidental. Una revisión sistemática. Medicina clinica, 156(9), 449–458. https://doi.org/10.1016/j.medcli.2020.11.009

[1.4] Greenberg, N., Weston, D., Hall, C., Caulfield, T., Williamson, V., & Fong, K. (2021). Mental health of staff working in intensive care during Covid-19. Occupational medicine (Oxford, England), 71(2), 62–67. https://doi.org/10.1093/occmed/kqaa220

[1.5] Bryson, E. O., & Silverstein, J. H. (2008). Addiction and substance abuse in anesthesiology. Anesthesiology, 109(5), 905–917. https://doi.org/10.1097/ALN.0b013e3181895bc1

[1.6] Baldisseri M. R. (2007). Impaired healthcare professional. Critical care medicine, 35(2 Suppl), S106–S116. https://doi.org/10.1097/01.CCM.0000252918.87746.96

[1.7] Rodríguez-Rey, R., Palacios, A., Alonso-Tapia, J., Pérez, E., Álvarez, E., Coca, A., Mencía, S., Marcos, A., Mayordomo-Colunga, J., Fernández, F., Gómez, F., Cruz, J., Ordóñez, O., & Llorente, A. (2019). Burnout and posttraumatic stress in paediatric critical care personnel: Prediction from resilience and coping styles. Australian critical care : official journal of the Confederation of Australian Critical Care Nurses, 32(1), 46–53. https://doi.org/10.1016/j.aucc.2018.02.003

[2.1] Stone, E. M., Kennedy-Hendricks, A., Barry, C. L., Bachhuber, M. A., & McGinty, E. E. (2021). The role of stigma in U.S. primary care physicians' treatment of opioid use disorder. Drug and alcohol dependence, 221, 108627. https://doi.org/10.1016/j.drugalcdep.2021.108627

[2.2] Latkin, C. A., Edwards, C., Davey-Rothwell, M. A., & Tobin, K. E. (2017). The relationship between social desirability bias and self-reports of health, substance use, and social network factors among urban substance users in Baltimore, Maryland. Addictive behaviors, 73, 133–136. https://doi.org/10.1016/j.addbeh.2017.05.005

[2.3] Welte, J. W., & Russell, M. (1993). Influence of socially desirable responding in a study of stress and substance abuse. Alcoholism, clinical and experimental research, 17(4), 758–761. https://doi.org/10.1111/j.1530-0277.1993.tb00836.x

[3.1]Maunder, R. G., Lancee, W. J., Balderson, K. E., Bennett, J. P., Borgundvaag, B., Evans, S., Fernandes, C. M., Goldbloom, D. S., Gupta, M., Hunter, J. J., McGillis Hall, L., Nagle, L. M., Pain, C., Peczeniuk, S. S., Raymond, G., Read, N., Rourke, S. B., Steinberg, R. J., Stewart, T. E., VanDeVelde-Coke, S., … Wasylenki, D. A. (2006). Long-term psychological and occupational effects of providing hospital healthcare during SARS outbreak. Emerging infectious diseases, 12(12), 1924–1932. https://doi.org/10.3201/eid1212.060584

[3.2]Roberts, N. J., Kelly, C. A., Lippiett, K. A., Ray, E., & Welch, L. (2021). Experiences of nurses caring for respiratory patients during the first wave of the COVID-19 pandemic: an online survey study. BMJ open respiratory research, 8(1), e000987. https://doi.org/10.1136/bmjresp-2021-000987

[3.3] What organisations around the world are doing to help improve doctors’ wellbeing

BMJ 2020; 369 doi: https://doi.org/10.1136/bmj.m1541

3. The appropriate statistical methods were chosen.

Author’s Reply: None.

4. The authors are unable to make the data publicly available due to its sensitive nature, but it is available upon reasonable request.

Author’s Reply: None.

5. The language errors make it difficult to understand at times, and the awkward expression of certain terms may cause readers to misunderstand what the authors are trying to convey.

e.g: 

"licensed from work" -- I believe the authors were trying to say that these healthcare workers were absent from work?

"Although most of them shared their homes with individuals with greater risk for severe infection, only 21% of them left home during the pandemic" -- It is unclear why or how these individuals are at greater risk of severe infection. Do you mean to say they belong to a vulnerable population (e.g chronic illnesses/elderly)? The term "left home" should be more appropriately expressed as "sought alternative accommodation away from home".

"However, with regards to the non-statistically significant associations, we believe they should be taken into account because of the biological plausibility they carry within them." -- Am not sure what biological plausibility means in this context

"Pharmaceuticals" -- I believe it should be "pharmacists"

Author’s Reply: We appreciate the suggestions made regarding the proper use of the English language. They were incorporated into the manuscript. Also, as an attempt to improve the quality of the text, two native English speaker biomedical scientists, by any means involved with this project, blindly and independently reviewed the text, and further suggestions were also incorporated to improve the overall quality of the text.

6. Other comments:

- The abstract should not contain abbreviations that have not been spelled out in full at first mention.

Author’s Reply: the abstract has been re-write without abbreviations and with a clearer language.

- Please provide a citation for the way the IES-R and ASSIST 2.0 scores were categorized. Are there scoring guidelines available?

Author’s Reply: 

#1 IES-R: the citation for the score categorization of the IES-R can be found below; note that we used the most recent version of the tool which was validated to the Brazilian population. In the manuscript we included the original citation as well as the citation for the validation study.

REFERENCE

[4.1] Weiss, D.S. (2007). The Impact of Event Scale: Revised. In J.P. Wilson & C.S. Tang (Eds.), Cross-cultural assessment of psychological trauma and PTSD (pp. 219-238). New York: Springer.

#2 ASSIST 2.0: the citation below refers to the article led by the World Health Organization (WHO), in which it is possible to find the interpretation for the scoring categories of the ASSIST tool. Similarly to the IES-R, we selected the most updated version of the ASSIST questionnaire which was validated to the Brazilian population.

REFERENCE

[4.2] WHO ASSIST Working Group (2002). The Alcohol, Smoking and Substance Involvement Screening Test (ASSIST): development, reliability and feasibility. Addiction (Abingdon, England), 97(9), 1183–1194. https://doi.org/10.1046/j.1360-0443.2002.00185.x

As suggested, both references were properly incorporated to the manuscript.

- Please refer to a standardized format (e.g APA) for the presentation of data in tables. The way it is currently organized can be confusing for readers. Also, all abbreviations in tables and its title (e.g ICU, IES-R) should be explained in the footnotes.

Author’s Reply: Tables were properly formatted according to the APA Publishing Manual (7th edition), and footnotes now describe all abbreviations used; also, explanations regarding the presented variables were provided whenever needed, to facilitate interpretation by readers. 

Comments - Reviewer #2 

The study is interesting but is single-centre. 

Author’s Reply: Dear Reviewer #2, thank you for your valuable comments! We agree with you in the sense that being a single-centre project is a limitation of the presenting study. Collaborations with other institutions were discussed, but due to logistical constraints, we could not perform a multi-centre project. Nevertheless, we believe that this study has been done over solid methodological basis, which may account for its internal validity. Also, although the studied population is from Brazil, our results are in accordance with results from studies conducted in different countries, which may indicate that the presenting study also has appreciable external validity.

There is no comparison between the different critical care professions.

Author’s Reply: We are glad you mentioned this issue, as this has been discussed within our research group. On the contrary of what we expected to see, there was no significant difference in the scores among the various critical care professions. We hypothesize that this may be due to lack of statistical power. Interestingly, the larger group in the study sample accounts for nurses and nursing assistants, which, again, is in accordance with which has been found by other similar studies conducted in other countries. Thus, the present study may help to demonstrate that larger sample sizes may be required to evaluate differences in the stress burden among the various professions involved in running an ICU; alternatively, the results presented in this study may also suggest that profession per se is not independently associated with higher risks of developing traumatic stress disorders and/or substance abuse.

The authors do not report anxiety or depression scores.

Author’s Reply: We appreciate your attention to this detail. Indeed, anxiety and depression have been recurrently reported as common mental health problems among ICU workers. During the development of the research question for this study, we conducted a thorough literature review, and we identified a significant gap in the knowledge regarding substance abuse and traumatic stress disorders in the studied population. Thus, although we were aware that depression and anxiety disorders could also be elevated in the studied population, we aimed to explore other severe comorbidities, trying to fill a specific knowledge gap previously identified by our literature review. 

Finally, the authors did not go into enough depth in their analyses.

Author’s Reply: Dear Reviewer #2, we are sorry to hear that. While planning this project, we tried to build a strong investigation, deep enough to contribute with our scientific community in this moment of collective efforts. We understand the limitations of the study, but we hope that, as a result of this manuscript, ICU workers can benefit from better institutional support. Nonetheless, we appreciate your comments and suggestions, and we believe they have helped to improve this manuscript.

Comments - Reviewer #3

The title is accurate or relevant. The aims of the study are clearly stated. The study is original. The study is useful and relevant to the aims of the Journal. The design of the study is appropriate. The sample size, selection and composition are appropriate. Methods used to collect data (e.g. validated questionnaires and instruments, observational techniques) are appropriate. Qualitative or quantitative methods used to analyse the data are appropriate. Details of the methods (including settings and locations, procedures, dates of recruitment and follow-up or main outcomes) are clearly reported. The data are less than 5 years old. The study was approved by a research ethics committee prior to data collection. Participants were asked for informed consent prior to data collection or informed consent was not required. The qualitative or quantitative analyses were applied appropriately. Missing data, e.g. non-respondents, drop-outs or non-responses, have been accounted for. The results are clearly presented and explained. No further qualitative or quantitative analysis is required. The authors reflect on the strengths and limitations of the study. The results are compared to related findings in the literature. The results are discussed in relation to the relevant research, practice or policy issues. The discussion and conclusions do not speculate beyond what has been shown in this study. The article has a logical construction in a suitable format. The article has an appropriate length (not unnecessarily long or too short to be useful). The writing is in a good standard of English, grammatically correct and easy to understand. The abstract is in an unstructured format and is sufficiently informative. Any tables and figures are all necessary, clearly annotated and easy to follow.

Author’s Reply: Dear Reviewer #3, we were extremely happy to receive your comments. Thank you for your time and consideration while evaluating our manuscript, we really appreciate it.

---

## [Decision Letter · Decision Letter 1]

14 Dec 2021

PONE-D-21-17265R1Stress and Substance Abuse among workers during the COVID-19 Pandemic in an Intensive Care Unit: a cross-sectional study.PLOS ONE

Dear Dr. Pestana,

Thank you for submitting your manuscript to PLOS ONE. After careful consideration, we feel that it has merit but does not fully meet PLOS ONE’s publication criteria as it currently stands. Therefore, we invite you to submit a revised version of the manuscript that addresses the points raised during the review process.

We look forward to receiving your revised manuscript.

Kind regards,

Sanjay Kumar Singh Patel, Ph.D.

Academic Editor

PLOS ONE

Journal Requirements:

Reviewers' comments:

Reviewer's Responses to Questions

**Comments to the Author**

1. If the authors have adequately addressed your comments raised in a previous round of review and you feel that this manuscript is now acceptable for publication, you may indicate that here to bypass the “Comments to the Author” section, enter your conflict of interest statement in the “Confidential to Editor” section, and submit your "Accept" recommendation.

Reviewer #1: (No Response)

Reviewer #2: All comments have been addressed

Reviewer #3: All comments have been addressed

2. Is the manuscript technically sound, and do the data support the conclusions?

Reviewer #1: Yes

Reviewer #2: Yes

Reviewer #3: Yes

3. Has the statistical analysis been performed appropriately and rigorously? 

Reviewer #1: Yes

Reviewer #2: Yes

Reviewer #3: Yes

4. Have the authors made all data underlying the findings in their manuscript fully available?

Reviewer #1: Yes

Reviewer #2: Yes

Reviewer #3: Yes

5. Is the manuscript presented in an intelligible fashion and written in standard English?

Reviewer #1: No

Reviewer #2: Yes

Reviewer #3: Yes

6. Review Comments to the Author

Reviewer #1: I thank the authors for careful consideration of my previous comments. I am largely satisfied with the revisions made, but still have a few areas of concern as highlighted below:

1. In the Results section, Paragraph on "Psychiatric Medical History", I believe the p-values reported are erroneous. It should be p = .034 and p = .044. Similarly, the p-values in Table 3.2 will need to be checked again to ensure that they correspond to the text.

2. There are still issues with the language in multiple areas, although I greatly appreciate the effort undertaken by the authors to improve the language of the manuscript thus far.

Some examples include:

- [Incomplete sentence] "All the professionals working at an oncological COVID-19 ICU, regardless of their role (health professionals and non-health professionals), who were on duty for at least during one shift through the period of July to October/2020."

- [Error indicated in caps] "Professionals with infected relatives .... significantly more likely to score higher in the IES-R, therefore more likely to suffer WITH PTSD or ASD."

- [Error indicated in caps] "There was an association trend between higher scores and those who had sought alternative accommodation AWAY FROM during the pandemic ...."

- [Error indicated in caps] "For cannabis, stimulants, and cocaine, higher scores had an ... direct EXPOSITION to infected patients."

- [Error indicated in caps] "It is possible that those results REFLEX the availability of those substances."

- ["Provided that" should be replaced with "GIVEN that some of those workers MAY HAVE HAD.."] Provided that some of those workers had psychological or psychiatric reasons for work leave, our findings can be underestimated.

- [Should be phrased as "study had no FOLLOW-UP sessions] "Fourthly, as a cross-sectional study, it was not possible to determine causal relationships nor risk factors and the participants were not followed."

3. Minor suggestion to change the header "ASSIST 2.0 Results" to "Associations with substance abuse" instead.

4. Please provide the citation for the cut-off points/categorization of IES and ASSIST 2.0 scores after listing the cut-offs in order to allow readers to easily search for the original paper that provides these cut-offs.

i.e "The IES-R scores were categorized as follows: 1-11 = few/no signs of ASD/PTSD; 12-32 = several signs of ASD/PTSD, patient monitoring is recommended; ≥33 = highly suggestive of ASD/PTSD, immediate psychiatric evaluation is recommended [INSERT CITATION HERE]."

5. Authors should consider elaborating a little on how their findings may be generalizable to ICUs outside Brazil. A suggestion is given below.

"the current results are in accordance those presented by other authors, in other populations of ICU workers (e.g., Netherlands [4], Canada [20], United States [21], and England [19]), suggesting that findings from this study are applicable and should be taken into consideration by professionals working in ICUs globally."

Reviewer #2: The authors have responded in a satisfactory manner to all comments.

I have no additional comments to make.

Reviewer #3: The design of the study is appropriate

The writing is in a good standard of English, grammatically correct and easy to understand

The results are clearly presented and explained

---

## [Author Response · Author response to Decision Letter 1]

24 Jan 2022

Response to the Editor and Reviewers

Dear Reviewers and Dr. Patel (Editorial Board of PLOS One). We are pleased to share with you our response to the comments made regarding our manuscript entitled “Stress and Substance Abuse among workers during the COVID-19 Pandemic in an Intensive Care Unit: a cross-sectional study”. We hope that our replies can properly address the issues highlighted by you. Thank you for your attention to this.

Sincerely,

Diego V. S. Pestana (on behalf of the authors)

Comments – Editor:

Author’s Reply: Dear Dr. Patel, thank you for your help with this manuscript. We reviewed the reference list completely and, although we could not find any retracted paper, we identified some points that could be optimized, as discussed below (between brackets we informed which action was made regarding each reference). Additionally, we checked all the references to make sure they were available online at their original publisher/journal website, without retraction notes. All references were written according to the Vancouver style, as indicated at PLOS One website.

Reference #7: Baldisseri MR. Impaired healthcare professional. Critical Care Medicine. 2007 Feb;35(Suppl):S106–16.

[Substitution]

Although the article appears after a simple search on PubMed, whenever we tried to access it using the publisher/journal website the link led us to a blank webpage. We searched for retraction notes or such type of publication regarding this paper but there was none. Therefore, we decided to look for a new reference article to substitute it. After reading several other papers, we chose the article below, which also supports what we wrote in the respective excerpt of the presenting manuscript.

“DeFord S, Bonom J, Durbin T. A review of literature on substance abuse among anaesthesia providers. Journal of Research in Nursing. 2019 Dec 22;24(8):587–600.”

Reference #15: Group WAW. The Alcohol, Smoking and Substance Involvement Screening Test (ASSIST): development, reliability and feasibility. Addiction. 2002 Sep;97(9):1183–94.

[Substitution]

Although the previous reference also contained information regarding the ASSIST 2.0 tool, interpreting scores based solely on this paper could be difficult for readers. The new reference refers to the manual more recently published on behalf of the World Health Organization and allows readers to quickly find the information to which our study refers. Therefore, we thought it should substitute the former reference.

“Humeniuk R, Ali R, Babor TF, Farrell M, Formigoni ML, Jittiwutikarn J, et al. Validation of the alcohol, smoking and substance involvement screening test (ASSIST). Addiction. 2008 Jun;103(6):1039–47.”

Reference #17: Weiss DS. The Impact of Event Scale: Revised. In: Wilson JP, Tang CS, editors. Cross-cultural assessment of psychological trauma and PTSD . New York: Springer; 2007. p. 219–38. 

[Correction] 

The first citation was incorrect. It improperly mixed information regarding the book and the chapter being cited. We corrected it with the proper citation, as listed below.

“Weiss DS. The Impact of Event Scale: Revised. In: Cross-Cultural Assessment of Psychological Trauma and PTSD. Boston, MA: Springer US; p. 219–38.”

Comments - Reviewer #1

I thank the authors for careful consideration of my previous comments. I am largely satisfied with the revisions made, but still have a few areas of concern as highlighted below:

1. In the Results section, Paragraph on "Psychiatric Medical History", I believe the p-values reported are erroneous. It should be p = .034 and p = .044. Similarly, the p-values in Table 3.2 will need to be checked again to ensure that they correspond to the text.

Author’s Reply: Dear Reviewer #1, thank you for your very constructive comments. After carefully reviewing the paragraph you mentioned, we identified a discordance between the p values presented in the paragraph and in Table 3.1. We reassessed our statistical tests and the correct values are p = 0.341 (history of psychiatric disease) and p = 0.44 (history of anxiety disorders). The values were properly corrected (please find our STATA output below, with the respective p values). Also, we corrected the tables’ titles according to the new order they were cited in the text. Former Table 3.1 was renamed to Table 3.2; former Table 3.2 was renamed to Table 3.1.

In this paragraph we highlighted some factors that had exhibited what we considered an association trend towards higher IES-R scores. Although they provided some interesting insights regarding the issues involved with stress and substance abuse among participants, those factors did not achieve statistical significance when considering p <0.05. Therefore, we believe that the terms used to describe those findings should be substituted: we chose “Factors without an association with IES-R scores”. Although the association between those factors and IES-R scores did not achieve statistical significance, the presenting results may still be interesting to readers, and future studies with more statistical power may show different results. Finally, we chose to present those elements in a different table (i.e., Table 3.1) to make sure readers would not confuse them with the ones that achieved statistical significance (i.e., Table 3.2).

Finally, we merged the subtopics “Personal factors” and “Occupational factors”, creating a new subtopic entitled “Personal and occupational factors”. We believe that this change made the excerpt conciser to eventual readers. 

We hope your concerns regarding this topic were properly and satisfactorily addressed. Thank you for your valuable suggestions!

2. There are still issues with the language in multiple areas, although I greatly appreciate the effort undertaken by the authors to improve the language of the manuscript thus far.

Author’s Reply: Again, we are pleased to fully accept your suggestions regarding English issues. Below you will find a list with corrections made after your suggestions, following the same order you presented them. We hope those changes shall address your concerns regarding this matter.

- [Incomplete sentence] "All the professionals working at an oncological COVID-19 ICU, regardless of their role (health professionals and non-health professionals), who were on duty for at least during one shift through the period of July to October/2020."

Author’s Reply: we added the excerpt “were invited to participate in this study” after the sentence mentioned.

- [Error indicated in caps] "Professionals with infected relatives .... significantly more likely to score higher in the IES-R, therefore more likely to suffer WITH PTSD or ASD."

Author’s Reply: we substituted the word “with” by “from” (i.e., to suffer from PTSD or ASD).

- [Error indicated in caps] "There was an association trend between higher scores and those who had sought alternative accommodation AWAY FROM during the pandemic ...."

Author’s Reply: we completed the sentence with the word “home” (i.e., away from home during the pandemic). To ensure this error wasn’t present in other similar sentences, we checked the other parts in which this term was used, and they were properly written.

- [Error indicated in caps] "For cannabis, stimulants, and cocaine, higher scores had an ... direct EXPOSITION to infected patients."

Author’s Reply: we substituted the word “exposition” by “exposure”. Like what we did in the previous suggestion, all other similar excerpts were checked for mistakes, and they were properly written.

- [Error indicated in caps] "It is possible that those results REFLEX the availability of those substances."

Author’s Reply: we substituted the word “reflex” by “reflected”. No other similar mistakes were found.

- ["Provided that" should be replaced with "GIVEN that some of those workers MAY HAVE HAD.."] Provided that some of those workers had psychological or psychiatric reasons for work leave, our findings can be underestimated.

Author’s Reply: we replaced the terms as mentioned above. 

- [Should be phrased as "study had no FOLLOW-UP sessions] "Fourthly, as a cross-sectional study, it was not possible to determine causal relationships nor risk factors and the participants were not followed."

Author’s Reply: we rephrased this excerpt as “Fourthly, this study had no follow-up sessions. Therefore, it was not possible to determine causal relationships or risk factors”.

3. Minor suggestion to change the header "ASSIST 2.0 Results" to "Associations with substance abuse" instead.

Author’s Reply: we changed the header as suggested; we agree that the suggested version is more suitable to readers. Thank you!

4. Please provide the citation for the cut-off points/categorization of IES and ASSIST 2.0 scores after listing the cut-offs in order to allow readers to easily search for the original paper that provides these cut-offs.

i.e "The IES-R scores were categorized as follows: 1-11 = few/no signs of ASD/PTSD; 12-32 = several signs of ASD/PTSD, patient monitoring is recommended; ≥33 = highly suggestive of ASD/PTSD, immediate psychiatric evaluation is recommended [INSERT CITATION HERE]."

Author’s Reply: we inserted the respective citations to the IES-R and ASSIST 2.0 scores whenever those categorizations were mentioned. 

5. Authors should consider elaborating a little on how their findings may be generalizable to ICUs outside Brazil. A suggestion is given below.

"the current results are in accordance those presented by other authors, in other populations of ICU workers (e.g., Netherlands [4], Canada [20], United States [21], and England [19]), suggesting that findings from this study are applicable and should be taken into consideration by professionals working in ICUs globally."

Author’s Reply: after reading your suggestion, we added the excerpt “Thus, it is likely that the findings of the presenting study could be generalizable to ICU workers from different populations and should be taken into consideration by professionals working globally”. We agree that such a phrase may clarify the main idea of this paragraph. Thank you!

Comments - Reviewer #2

The authors have responded in a satisfactory manner to all comments. I have no additional comments to make.

Author’s Reply: We appreciate your attention when reviewing this manuscript. We hope the corrections made after your previous comments had satisfactorily addressed your concerns and improved the quality of the current manuscript.

Comments - Reviewer #3

The design of the study is appropriate. The writing is in a good standard of English, grammatically correct and easy to understand. The results are clearly presented and explained.

Author’s Reply: We are glad to read your comments to this manuscript, thank you for your help during this review process.

---

## [Editor Report · Decision Letter 2]

31 Jan 2022

Stress and Substance Abuse among workers during the COVID-19 Pandemic in an Intensive Care Unit: a cross-sectional study.

PONE-D-21-17265R2

Dear Dr. Pestana,

We’re pleased to inform you that your manuscript has been judged scientifically suitable for publication and will be formally accepted for publication once it meets all outstanding technical requirements.

Kind regards,

Sanjay Kumar Singh Patel, Ph.D.

Academic Editor

PLOS ONE

---

## [Editor Report · Acceptance letter]

3 Feb 2022

PONE-D-21-17265R2 

Stress and substance abuse among workers during the COVID-19 pandemic in an intensive care unit: a cross-sectional study. 

Dear Dr. Vinicius Santinelli Pestana:

I'm pleased to inform you that your manuscript has been deemed suitable for publication in PLOS ONE. Congratulations! Your manuscript is now with our production department. 

Kind regards, 

on behalf of

Dr. Sanjay Kumar Singh Patel 

Academic Editor

PLOS ONE